# GROWING EFFICIENT DEEP NETWORKS BY STRUCTURED CONTINUOUS SPARSIFICATION

**Xin Yuan**
University of Chicago
yuanx@uchicago.edu

**Pedro Savarese**
TTI-Chicago
savarese@ttic.edu

**Michael Maire**
University of Chicago
mmaire@uchicago.edu

## ABSTRACT

We develop an approach to growing deep network architectures over the course of training, driven by a principled combination of accuracy and sparsity objectives. Unlike existing pruning or architecture search techniques that operate on full-sized models or supernet architectures, our method can start from a small, simple seed architecture and dynamically grow and prune both layers and filters. By combining a continuous relaxation of discrete network structure optimization with a scheme for sampling sparse subnetworks, we produce compact, pruned networks, while also drastically reducing the computational expense of training. For example, we achieve $49.7\%$ inference FLOPs and $47.4\%$ training FLOPs savings compared to a baseline ResNet-50 on ImageNet, while maintaining $75.2\%$ top-1 accuracy — all without any dedicated fine-tuning stage. Experiments across CIFAR, ImageNet, PASCAL VOC, and Penn Treebank, with convolutional networks for image classification and semantic segmentation, and recurrent networks for language modeling, demonstrate that we both train faster and produce more efficient networks than competing architecture pruning or search methods.

## 1 INTRODUCTION

Deep neural networks are the dominant approach to a variety of machine learning tasks, including image classification (Krizhevsky et al., 2012; Simonyan & Zisserman, 2015), object detection (Girshick, 2015; Liu et al., 2016), semantic segmentation (Long et al., 2015; Chen et al., 2017) and language modeling (Zaremba et al., 2014; Vaswani et al., 2017; Devlin et al., 2019). Modern neural networks are overparameterized and training larger networks usually yields improved generalization accuracy. Recent work (He et al., 2016; Zagoruyko & Komodakis, 2016; Huang et al., 2017) illustrates this trend through increasing *depth* and *width* of convolutional neural networks (CNNs). Yet, training is compute-intensive, and real-world deployments are often limited by parameter and compute budgets.

Neural architecture search (NAS) (Zoph & Le, 2017; Liu et al., 2019; Luo et al., 2018; Pham et al., 2018; Savarese & Maire, 2019) and model pruning (Han et al., 2016; 2015; Guo et al., 2016) methods aim to reduce these burdens. NAS addresses an issue that further compounds training cost: the enormous space of possible network architectures. While hand-tuning architectural details, such as the connection structure of convolutional layers, can improve performance (Iandola et al., 2016; Sifre & Mallat, 2014; Chollet, 2017; Howard et al., 2017; Zhang et al., 2018; Huang et al., 2018), a principled way of deriving such designs remains elusive. NAS methods aim to automate exploration of possible architectures, producing an efficient design for a target task under practical resource constraints. However, during training, most NAS methods operate on a large *supernet* architecture, which encompasses candidate components beyond those that are eventually selected for inclusion in the resulting network (Zoph & Le, 2017; Liu et al., 2019; Luo et al., 2018; Pham et al., 2018; Savarese & Maire, 2019). Consequently, NAS-based training may typically be more thorough, but more computationally expensive, than training a single hand-designed architecture.

Model pruning techniques similarly focus on improving the resource efficiency of neural networks during inference, at the possible expense of increased training cost. Common strategies aim to generate a lighter version of a given network architecture by removing individual weights (Han et al., 2015; 2016; Molchanov et al., 2017) or structured parameter sets (Li et al., 2017; He et al., 2018; Luo et al., 2017). However, the majority of these methods train a full-sized model prior to pruning and,

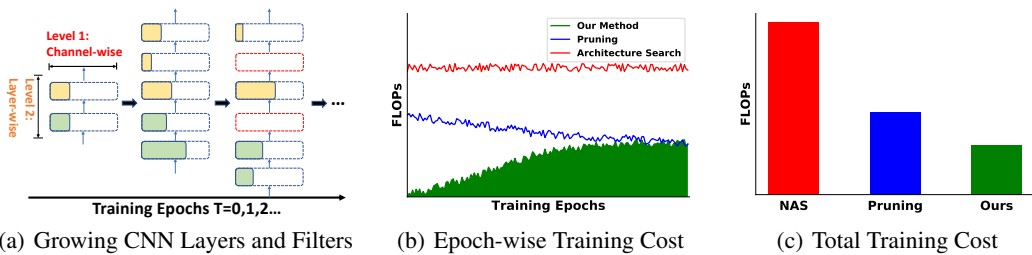

(a) Growing CNN Layers and Filters    (b) Epoch-wise Training Cost    (c) Total Training Cost

Figure 1: **Growing Networks during Training.** We define an architecture configuration space and simultaneously adapt network structure and weights. **(a)** Applying our approach to CNNs, we maintain auxiliary variables that determine how to grow and prune both filters (*i.e.* channel-wise) and layers, subject to practical resource constraints. **(b)** By starting with a small network and growing its size, we utilize fewer resources in early training epochs, compared to pruning or NAS methods. **(c)** Consequently, our method significantly reduces the total computational cost of training, while delivering trained networks of comparable or better size and accuracy.

after pruning, utilize additional fine-tuning phases in order to maintain accuracy. Hubara et al. (2016) and Rastegari et al. (2016) propose the use of binary weights and activations, allowing inference to benefit from reduced storage costs and efficient computation through bit-counting operations. Yet, training still involves tracking high-precision weights alongside lower-precision approximations.

We take a unified view of pruning and architecture search, regarding both as acting on a configuration space, and propose a method to dynamically grow deep networks by continuously reconfiguring their architecture during training. Our approach not only produces models with efficient inference characteristics, but also reduces the computational cost of training; see Figure 1. Rather than starting with a full-sized network or a supernet, we start from simple seed networks and progressively adjust (grown and prune) them. Specifically, we parameterize an architectural configuration space with indicator variables governing addition or removal of structural components. Figure 2(a) shows an example, in the form of a two-level configuration space for CNN layers and filters. We enable learning of indicator values (and thereby, architectural structure) via combining a continuous relaxation with binary sampling, as illustrated in Figure 2(b). A per-component temperature parameter ensures that long-lived structures are eventually baked into the network's discrete architectural configuration.

While the recently proposed AutoGrow (Wen et al., 2020) also seeks to grow networks over the course of training, our technical approach differs substantially and leads to significant practical advantages. At a technical level, AutoGrow implements an architecture search procedure over a predefined modular structure, subject to hand-crafted, accuracy-driven growing and stopping policies. In contrast, we parameterize architectural configurations and utilize stochastic gradient descent to learn the auxiliary variables that specify structural components, while simultaneously training the weights within those components. Our unique technical approach yields the following advantages:

- **Fast Training by Growing:** Training is a unified procedure, from which one can request a network structure and associated weights at any time. Unlike AutoGrow and the majority of pruning techniques, fine-tuning to optimize weights in a discovered architecture is optional. We achieve excellent results even without any fine-tuning stage.

- **Principled Approach via Learning by Continuation + Sampling:** We formulate our approach in the spirit of learning by continuation methods, which relax a discrete optimization problem to an increasingly stiff continuous approximation. Critically, we introduce an additional sampling step to this strategy. From this combination, we gain the flexibility of exploring a supernet architecture, but the computational efficiency of only actually training a much smaller active subnetwork.

- **Budget-Aware Optimization Objectives:** The parameters governing our architectural configuration are themselves updated via gradient decent. We have flexibility to formulate a variety of resource-sensitive losses, such as counting total FLOPs, in terms of these parameters.

- **Broad Applicability:** Though we use progressive growth of CNNs in width and depth as a motivating example, our technique applies to virtually any neural architecture. One has flexibility in how to parameterize the architecture configuration space. We also show results with LSTMs.

We demonstrate these advantages while comparing to recent NAS and pruning methods through extensive experiments on classification, semantic segmentation, and word-level language modeling.

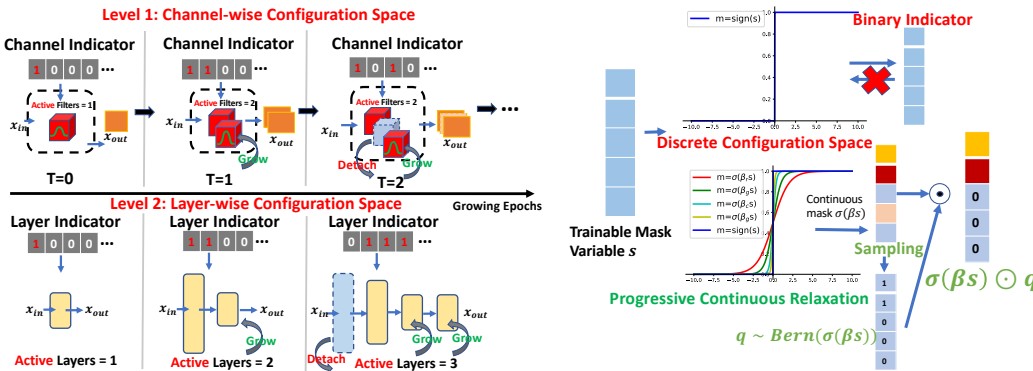

(a) Architectural Configuration Space for CNNs      (b) Optimization with Structured Continuation

Figure 2: **Technical Framework.** **(a)** We periodically restructure a CNN by querying binary indicators that define a two-level configuration space for filters and layers. **(b)** To make optimization feasible while growing networks, we derive these binary indicators from trainable continuous mask variables. We employ a structured extension of continuous sparsification (Savarese et al., 2020), combined with sampling. Binary stochastic auxiliary variables $q$, sampled according to $\sigma(\beta s)$, generate the discrete components active at a particular time.

## 2 RELATED WORK

**Network Pruning.** Pruning methods can be split into two groups: those pruning individual weights and those pruning structured components. Individual weight-based pruning methods vary on the removal criteria. For example, Han et al. (2015) propose to prune network weights with small magnitude, and subsequently quantize those remaining (Han et al., 2016). Louizos et al. (2018) learn sparse networks by approximating $\ell_0$-regularization with a stochastic reparameterization. However, sparse weights alone often only lead to speedups on dedicated hardware with supporting libraries.

In structured methods, pruning is applied at the level of neurons, channels, or even layers. For example, L1-pruning (Li et al., 2017) removes channels based on the norm of their filters. He et al. (2018) use group sparsity to smooth the pruning process after training. MorphNet (Gordon et al., 2018) regularizes weights towards zero until they are small enough such that the corresponding output channels are marked for removal from the network. Intrinsic Structured Sparsity (ISS) (Wen et al., 2018) works on LSTMs (Hochreiter & Schmidhuber, 1997) by collectively removing the columns and rows of the weight matrices via group LASSO. Although structured pruning methods and our algorithm share the same spirit of generating efficient models, we gain training cost savings by growing networks from small initial architectures instead of pruning full-sized ones.

**Neural Architecture Search.** NAS methods have greatly improved the performance achieved by small network models. Pioneering NAS approaches use reinforcement learning (Zoph et al., 2018; Zoph & Le, 2017) and genetic algorithms (Real et al., 2019; Xie & Yuille, 2017) to search for transferable network blocks whose performance surpasses many manually designed ones. However, such approaches require massive computation during the search — typically thousands of GPU days. To reduce computational cost, recent efforts utilize more efficient search techniques, such as direct gradient-based optimization (Liu et al., 2019; Luo et al., 2018; Pham et al., 2018; Tan et al., 2019; Cai et al., 2019; Wortsman et al., 2019). Nevertheless, most NAS methods perform search in a supernet space which requires more computation than training typically-sized architectures.

**Network Growing.** Network Morphism (Wei et al., 2016) searches for efficient deep networks by extending layers while preserving the parameters. Recently proposed Autogrow (Wen et al., 2020) takes an AutoML approach to growing layers. These methods either require a specially-crafted policy to stop growth (*e.g.,* after a fixed number of layers) or rely on evaluating accuracy during training, incurring significant additional computational cost.

**Learning by Continuation.** Continuation methods are commonly used to approximate intractable optimization problems by gradually increasing the difficulty of the underlying objective, for example by adopting gradual relaxations to binary problems. Wu et al. (2019); Xie et al. (2019b; 2020) use gumbel-softmax (Jang et al., 2017) to back-propagate errors during architecture search and spatial feature sparsification. Savarese et al. (2020) propose *continuous sparsification* to speed up pruning and ticket search (Frankle & Carbin, 2019). Despite the success of continuation methods in producing sparse networks upon the completion of training, they do not operate on sparse networks

during training and instead work with a real-valued relaxation. Postponing actual elimination of near zeroed-out components prevents naive application of these methods from reducing training costs.

# 3 METHOD

## 3.1 ARCHITECTURAL CONFIGURATION SPACE

A network topology can be seen as a directed acyclic graph consisting of an ordered sequence of nodes. Each node $x_{in}^{(i)}$ is an input feature and each edge is a computation cell with *structured* hyperparameters (*e.g.,* filter and layer numbers in convolutional networks). An architectural configuration space can be parameterized by associating a mask variable $m \in \{0, 1\}$ with each computation cell (edge), which enables training-time pruning ($m = 1 \rightarrow 0$) and growing ($m = 0 \rightarrow 1$) dynamics.

As a running example, we consider a two-level configuration space for CNN architectures, depicted in Figure 2(a), that enables dynamically growing networks in both width (channel-wise) and depth (layer-wise). Alternative configuration spaces are possible; we defer to the Appendix details on how we parameterize the design of LSTM architectures.

**CNN Channel Configuration Space:** For a convolutional layer with $l_{in}$ input channels, $l_{out}$ output channels (filters) and $k \times k$ sized kernels, the $i$-th output feature is computed based on the $i$-th filter, *i.e.* for $i \in \{1, \ldots, l_{out}\}$:

$$x_{out}^{(i)} = f(x_{in}, \mathcal{F}^{(i)} \cdot m_c^{(i)}), \tag{1}$$

where $m_c^{(i)} \in \{0, 1\}$ is a binary parameter that removes the $i$-th output channel when set to zero and $f$ denotes the convolutional operation. $m_c^{(i)}$ is shared across a filter and broadcasts to the same shape as the filter tensor $\mathcal{F}^{(i)}$, enabling growing/pruning of the entire filter. As Figure 2(a) (top) shows, we start from a *slim* channel configuration. We then query the indicator variables and perform *state transitions*: (1) When flipping an indicator variable from 0 to 1 for the first time, we grow a randomly initialized filter and concatenate it to the network. (2) If an indicator flips from 1 to 0, we temporarily detach the corresponding filter from the computational graph; it will be grown back to the its original position if its indicator flips back to 1, or otherwise be permanently pruned at the end of training. (3) For other cases, the corresponding filters either survive and continue training or remain detached pending the next query to their indicators. Our method automates architecture evolution, provided we can train the indicators.

**CNN Layer Configuration Space:** To grow network depth, we design a layer configuration space in which an initial shallow network will progressively expand into a deep trained model, as shown in Figure 2(a) (bottom). Similar to channel configuration space, where filters serve as basic structural units, we require a unified formulation to support the growing of popular networks with shortcut connections (*e.g.,* ResNets) and without (*e.g.,* VGG-like plain nets). We first introduce an abstract layer class $f_{layer}$ as a basic structural unit, which operates on input features $x_{in}$ and generates output features $x_{out}$. $f_{layer}$ can be instantiated as convolutional layers for plain nets or residual blocks for ResNets, respectively. We define the layer configuration space as:

$$x_{out} = g(x_{in}; f_{layer} \cdot m_l^{(j)}) = \begin{cases} f_{layer}(x_{in}), & \text{if } m_l^{(j)} = 1 \\ x_{in}, & \text{if } m_l^{(j)} = 0 \end{cases}, \tag{2}$$

where $m_l^{(j)} \in \{0, 1\}$ is the binary indicator for $j$-th layer $f_{layer}$, with which we perform state transitions analogous to the channel configuration space. Layer indicators have priority over channel indicators: if $m_l^{(j)}$ is set as 0, all filters contained in the corresponding layer will be detached, regardless of the state their indicators. We do not detach layers that perform changes in resolution (*e.g.,* strided convolution).

## 3.2 GROWING WITH STRUCTURED CONTINUOUS SPARSIFICATION

We can optimize a trade-off between accuracy and structured sparsity by considering the objective:

$$\min_{w, m_{c,l}, f_{layer}} L_E(g(f(x; w \odot m_c); f_{layer} \cdot m_l)) + \lambda_1 \|m_c\|_0 + \lambda_2 \|m_l\|_0, \tag{3}$$

where $f$ is the operation in Eq. (1) or Eq. (9) (in Appendix A.6), while $g$ is defined in Eq. (2). $\boldsymbol{w} \odot \boldsymbol{m}_c$ and $f_{layer} \cdot \boldsymbol{m}_l$ are general expressions of structured sparsified filters and layers and $L_E$ denotes a loss function (*e.g.,* cross-entropy loss for classification). The $\ell_0$ terms encourage sparsity, while $\lambda_{1,2}$ are trade-off parameters between $L_E$ and the $\ell_0$ penalties.

**Budget-aware Growing.** In practice, utilizing Eq. (3) might require a grid search on $\lambda_1$ and $\lambda_2$ until a network with desired sparsity is produced. To avoid such a costly procedure, we propose a budget-aware growing process, guided by a target budget in terms of model parameters or FLOPs. Instead of treating $\lambda_1$ and $\lambda_2$ as constants, we periodically update them as:

$$\lambda_1 \leftarrow \lambda_1^{\text{base}} \cdot \Delta u, \lambda_2 \leftarrow \lambda_2^{\text{base}} \cdot \Delta u, \quad (4)$$

where $\Delta u$ is calculated as the target sparsity $u$ minus current network sparsity $u_G$, and $\lambda_1^{\text{base}}$, $\lambda_2^{\text{base}}$ are initial base constants. In early growing stages, since the network is too sparse and $\Delta u$ is negative, the optimizer will drive the network towards a state with more capacity (wider/deeper). The regularization effect gradually weakens as the network's sparsity approaches the budget (and $\Delta u$ approaches zero). This allows us to adaptively grow the network and automatically adjust its sparsity level while simultaneously

---

**Algorithm 1** : Optimization

**Input:** Data $\boldsymbol{X} = (\boldsymbol{x}_i)_{i=1}^n$, labels $\boldsymbol{Y} = (\boldsymbol{y}_i)_{i=1}^n$
**Output:** Grown efficient model $G$
Initialize: $G$, $w$, $u$, $\lambda_1^{\text{base}}$ and $\lambda_2^{\text{base}}$.
Set $\boldsymbol{t}_s$ as all 0 vectors associating $\sigma$ functions.
**for** epoch $= 1$ **to** $T$ **do**
    Evaluate $G$'s sparsity $u_G$ and calculate
        $\Delta u = u - u_G$
    Update $\lambda_1 \leftarrow \lambda_1^{\text{base}} \cdot \Delta u$; $\lambda_2 \leftarrow \lambda_2^{\text{base}} \cdot \Delta u$
        in Eq. (6) using Eq. (4)
    **for** $r = 1$ **to** $R$ **do**
        Sample mini-batch $x_i, y_i$ from $\boldsymbol{X}, \boldsymbol{Y}$
        Train $G$ using Eq. (6) with SGD
    **end for**
    Sample indicators $q_{c,l} \sim \text{Bern}(\sigma(\beta s_{c,l}))$
    and record the index $idx$ where $q$ value is 1.
    Update $\boldsymbol{t}_s[idx] = \boldsymbol{t}_s[idx] + 1$
    Update $\boldsymbol{\beta}$ using Eq. (7)
**end for**
return G

---

training model weights. Appendix A.1 provides more detailed analysis. Our experiments default to defining budget by parameter count, but also investigate alternative notions of budget.

**Learning by Continuation.** Another issue in optimizing Eq. (3) is that $\|m_c\|_0$ and $\|m_l\|_0$ make the problem computationally intractable due to the combinatorial nature of binary states. To make the configuration space continuous and the optimization feasible, we borrow the concept of learning by continuation (Cao et al., 2017; Wu et al., 2019; Savarese et al., 2020; Xie et al., 2020). We reparameterize $m$ as the binary sign of a continuous variable $s$: $\text{sign}(s)$ is 1 if $s > 0$ and 0 if $s < 0$. We rewrite the objective in Eq. (3) as:

$$\min_{\boldsymbol{w},\boldsymbol{s}_{c,l}\neq 0,f_{layer}} L_E\Big(g\big(f(\boldsymbol{x};\boldsymbol{w} \odot \text{sign}(\boldsymbol{s}_c)); f_{layer} \cdot \text{sign}(\boldsymbol{s}_l)\big)\Big) + \lambda_1 \|\text{sign}(\boldsymbol{s}_c)\|_1 + \lambda_2 \|\text{sign}(\boldsymbol{s}_l)\|_1 . \quad (5)$$

We attack the hard and discontinuous optimization problem in Eq. (5) by starting with an *easier* objective which becomes *harder* as training proceeds. We use a sequence of functions whose limit is the sign operation: for any $s \neq 0$, $\lim_{\beta \to \infty} \sigma(\beta s) = \text{sign}(s)$ if $\sigma$ is sigmoid function or $\lim_{\beta \to 0} \sigma(\beta s) = \text{sign}(s)$ if $\sigma$ is gumbel-softmax $\frac{exp((-log(s_0)+g_1(s))/\beta)}{\sum_{j\in\{0,1\}} exp((-log(s_j)+g_j(s))/\beta)}$ (Jang et al., 2017), where $\beta > 0$ is a temperature parameter and $g_{0,1}$ is gumbel. By periodically changing $\beta$, $\sigma(\beta s)$ becomes harder to optimize, while the objectives converges to original discrete one.

**Maintaining Any-time Sparsification.** Although continuation methods can make the optimization feasible, they only conduct sparsification via a thresholding criterion in the inference phase. In this case, the train-time architecture is dense and not appropriate in the context of growing a network. To effectively reduce computational cost of training, we maintain a sparse architecture by introducing an 0-1 sampled auxiliary variable $q$ based on the probability value $\sigma(\beta s)$. Our final objective becomes:

$$\min_{\boldsymbol{w},\boldsymbol{s}_{c,l}\neq 0,f_{layer}} L_E\Big(g\big(f(\boldsymbol{x};\boldsymbol{w} \odot \sigma(\beta\boldsymbol{s}_c) \odot \boldsymbol{q}_c); f_{layer} \cdot \sigma(\beta\boldsymbol{s}_l) \cdot \boldsymbol{q}_l\big)\Big) + \lambda_1 \|\sigma(\beta\boldsymbol{s}_c)\|_1 + \lambda_2 \|\sigma(\beta\boldsymbol{s}_l)\|_1 , \quad (6)$$

where $\boldsymbol{q}_c$ and $\boldsymbol{q}_l$ are random variables sampled from $\text{Bern}(\sigma(\beta\boldsymbol{s}_c))$ and $\text{Bern}(\sigma(\beta\boldsymbol{s}_l))$, which effectively maintains any-time sparsification and avoids sub-optimal thresholding, as shown in Figure 2(b).

**Improved Temperature Scheduler.** In existing continuation methods, the initial $\beta$ value is usually set as $\beta_0 = 1$ and a scheduler is used at the end of each training epoch to update $\beta$ in all activation functions $\sigma$, typically following $\beta = \beta_0 \cdot \gamma^t$, where $t$ is current epoch and $\gamma$ is a hyperparameter ($> 1$ when $\sigma$ is the sigmoid function, $< 1$ when $\sigma$ is gumbel softmax). Both $\gamma$ and $t$ control the speed

Table 1: Comparison with the channel pruning methods L1-Pruning (Li et al., 2017), SoftNet (He et al., 2018), ThiNet (Luo et al., 2017), Provable (Liebenwein et al., 2020) and BAR (Lemaire et al., 2019) on CIFAR-10.

| Model | Method | Val Acc(%) | Params(M) | FLOPs(%) | Train-Cost Savings(×) |
|---|---|---|---|---|---|
| VGG -16 | Original | 92.9 ± 0.16 (-0.0) | 14.99 (100%) | 100 | 1.0× |
| | L1-Pruning | 91.8 ± 0.12 (-1.1) | 2.98 (19.9%) | 19.9 | 2.5× |
| | SoftNet | 92.1 ± 0.09 (-0.8) | 5.40 (36.0%) | 36.1 | 1.6× |
| | ThiNet | 90.8 ± 0.11 (-2.1) | 5.40 (36.0%) | 36.1 | 1.6× |
| | Provable | 92.4 ± 0.12 (-0.5) | 0.85 (5.7%) | 15.0 | 3.5× |
| | Ours | **92.50 ± 0.10 (-0.4)** | **0.754 ± 0.005 (5.0%)** | **13.55 ± 0.03** | **4.95 ± 0.17 ×** |
| ResNet -20 | Original | 91.3 ± 0.12 (-0.0) | 0.27 (100%) | 100 | 1.0× |
| | L1-Pruning | 90.9 ± 0.10 (-0.4) | 0.15 (55.6%) | 55.4 | 1.1× |
| | SoftNet | 90.8 ± 0.13 (-0.5) | 0.14 (53.6%) | 50.6 | 1.2× |
| | ThiNet | 89.2 ± 0.18 (-2.1) | 0.18 (67.1%) | 67.3 | 1.1× |
| | Provable | 90.8 ± 0.08 (-0.5) | 0.10 (37.3%) | 54.5 | 1.7× |
| | Ours | **90.91 ± 0.07 (-0.4)** | **0.096 ± 0.002 (35.8%)** | **50.20 ± 0.01** | **2.40 ± 0.09 ×** |
| WRN -28 -10 | Original | 96.2 ± 0.10 (-0.0) | 36.5 (100%) | 100 | 1.0× |
| | L1-Pruning | 95.2 ± 0.10 (-1.0) | 7.6 (20.8%) | 49.5 | 1.5× |
| | BAR(16x V) | 92.0 ± 0.08 (-4.2) | **2.3 (6.3%)** | **1.5** | 2.6× |
| | Ours | **95.32 ± 0.11 (-0.9)** | 3.443 ± 0.010 (9.3%) | 28.25 ± 0.04 | **3.12 ± 0.11×** |

at which the temperature increases during training. Continuation methods with global temperature schedulers have been successfully applied in pruning and NAS. However, in our case, a global schedule leads to unbalanced dynamics between variables with low and high sampling probabilities: increasing the temperature of those less sampled at early stages may hinder their training altogether, as towards the end of training the optimization difficulty is higher. To overcome this issue, we propose a separate, structure-wise temperature scheduler by making a simple modification: for each mask variable, instead of using the current epoch number $t$ to compute its temperature, we set a separate counter $t_s$ which is increased only when its associated indicator variable is sampled as 1 in Eq. (6). We define our structure-wise temperature scheduler as

$$\boldsymbol{\beta} = \beta_0 \cdot \gamma^{\boldsymbol{t_s}}, \tag{7}$$

where $\boldsymbol{t}_s$ are vectors associated with the $\sigma$ functions. Experiments use this separate scheduler by default, but also compare the two alternatives. Algorithm 1 summarizes our optimization procedure.

## 4 EXPERIMENTS

We evaluate our method against existing channel pruning, network growing, and neural architecture search (NAS) methods on: CIFAR-10 (Krizhevsky et al., 2014) and ImageNet (Deng et al., 2009) for image classification, PASCAL (Everingham et al., 2015) for semantic segmentation and the Penn Treebank (PTB) (Marcus et al., 1993) for language modeling. See dataset details in Appendix A.2. In tables, best results are highlighted in bold and second best are underlined.

### 4.1 COMPARING WITH CHANNEL PRUNING METHODS

**Implementation Details.** For fair comparison, we only grow filters while keeping other structured parameters of the network (number of layers/blocks) the same as unpruned baseline models. Our method involves two kinds of trainable variables: model weights and mask weights. For model weights, we adopt the same hyperparameters used to train the corresponding unpruned baseline models, except for setting the dropout keep probability for language modeling to 0.65. We initialize mask weights such that a single filter is activated in each layer. We train with SGD, an initial learning rate of 0.1, weight decay of $10^{-6}$ and momentum 0.9. Trade-off parameter $\lambda_1^{\text{base}}$ is set to 0.5 on all tasks; $\lambda_2$ is not used since we do not perform layer growing here. We set $\sigma$ as the sigmoid function and $\gamma$ as $100^{\frac{1}{T}}$ where $T$ is the total number of epochs.

**VGG-16, ResNet-20, and WideResNet-28-10 on CIFAR-10.** Table 1 summarizes the models produced by our method and competing channel pruning approaches. Note that training cost is calculated based on overall FLOPs during pruning and growing stages. Our method produces sparser networks with less accuracy degradation, and, consistently saves more computation during training — a consequence of growing from a simple network. For a aggressively pruned WideResNet-28-10, we observe that BAR (Lemaire et al., 2019) might not have enough capacity to achieve negligible accuracy drop, even with knowledge distillation (Hinton et al., 2015) during training. Note that we

Table 2: Comparison with channel pruning methods: L1-Pruning (Li et al., 2017), SoftNet (He et al., 2018) and Provable (Liebenwein et al., 2020) on ImageNet.

| Model | Method | Top-1 Acc(%) | Params(M) | FLOPs(%) | Train-Cost Savings(×) |
|---|---|---|---|---|---|
| ResNet-50 | Original | 76.1 (-0.0) | 23.0 (100%) | 100 | 1.0(×) |
| | L1-Pruning | 74.7 (-1.4) | 19.6 (85.2%) | 77.5 | 1.1(×) |
| | SoftNet | 74.6 (-1.5) | N/A | 58.2 | 1.2(×) |
| | Provable | **75.2 (-0.9)** | 15.2 (65.9%) | 70.0 | 1.2(×) |
| | Ours | **75.2 (-0.9)** | **14.1 (61.2%)** | **50.3** | **1.9**(×) |

Table 3: Results comparing with AutoGrow (Wen et al., 2020) on CIFAR-10 and ImageNet.

| Dataset | Methods | Variants | Found Net | Val Acc(%) | Depth | Sparse Channel |
|---|---|---|---|---|---|---|
| CIFAR-10 | Ours | *Basic3ResNet* | 23-29-31 | **94.50** | **83** | ✓ |
| | | *Plain3Net* | 11-14-19 | **90.99** | **44** | ✓ |
| | AutoGrow | *Basic3ResNet* | 42-42-42 | 94.27 | 126 | ✗ |
| | | *Plain3Net* | 23-22-22 | 90.82 | 67 | ✗ |
| ImageNet | Ours | *Bottleneck4ResNet* | 5-6-5-7 | **77.41** | **23** | ✓ |
| | | *Plain4Net* | 3-4-4-5 | **70.79** | **16** | ✓ |
| | AutoGrow | *Bottleneck4ResNet* | 6-7-3-9 | 77.33 | 25 | ✗ |
| | | *Plain4Net* | 5-5-5-4 | 70.54 | 19 | ✗ |

report our method's performance as mean ± standard deviation, computed over 5 runs with different random seeds. The small observed variance shows that our method performs consistently across runs.

**ResNet-50 and MobileNetV1 on ImageNet.** To validate effectiveness on large-scale datasets, we grow, from scratch, filters of the widely used ResNet-50 on ImageNet; we do not fine-tune. Table 2 shows our results best those directly reported in papers of respective competing methods. Our approach achieves 49.7% inference and 47.4% training cost savings in terms of FLOPs while maintaining 75.2% top-1 accuracy, without any fine-tuning stage. Appendix A.4 shows our improvements on the challenging task of growing channels of an already compact MobileNetV1. In addition, Figure 3 shows the top-1 accuracy/FlOPs trade-offs for MobileNetV1 on ImageNet, demonstrating that our method dominates competing approaches.

**Deeplab-v3-ResNet-101 on PASCAL VOC.** Appendix A.5 provides semantic segmentation results.

**2-Stacked-LSTMs on PTB:** We detail the extensions to recurrent cells and compare our proposed method with ISS based on vanilla two-layer stacked LSTM in Appendix A.6. As shown in Table 8, our method finds more compact model structure with lower training cost, while achieving similar perplexity on both validation and test sets.

## 4.2 COMPARING WITH AUTOGROW

**Implementation Details.** We grow both filters and layers. We follow AutoGrow's settings in exploring architectural variations that define our initial seed network, layer-wise configuration space and basic structural units $f_{layer}$: *Basic3ResNet*, *Bottleneck4ResNet*, *Plain3Net*, *Plain4Net*. Different from the initialization of AutoGrow using full-sized filters in each layer, our channel-wise configuration space starts from single filter and expands simultaneously with layers. Appendix A.7 contains a detailed comparison of seed architectures. For training model weights, we adopt the hyperparameters of the ResNet or VGG models corresponding to initial seed variants. For layer-wise and channel-wise mask variables, we initialize the weights such that only a single filter in each layer and one basic unit in each stage (*e.g.,* BasicBlock in *Basic3ResNet*) is active. We use SGD training with initial learning rate of 0.1, weight decay of $10^{-6}$ and momentum of 0.9 on all datasets. The learning rate scheduler is the same as for the corresponding model weights. Trade-off parameters $\lambda_1^{\text{base}}$ and $\lambda_2^{\text{base}}$ are set as 1.0 and 0.1 on all datasets. For fair comparison, we fine-tune our final models with 40 epochs and 20 epochs on CIFAR-10 and ImageNet, respectively.

**Results on CIFAR-10 and ImageNet.** Table 3 compares our results to those of AutoGrow. For all layer-wise growing variants across both datasets, our method consistently yields a better depth and width configuration than AutoGrow, in terms of accuracy and training/inference costs trade-off. Regarding the training time of *Bottleneck4ResNet* on ImageNet, AutoGrow requires 61.6 hours for the growing phase and 78.6 hours for fine-tuning on 4 TITAN V GPUs, while our method takes 48.2 and 31.3 hours, respectively. Our method offers 43% more train-time savings than AutoGrow. We not only require fewer training epochs, but also grow from a single filter to a relatively sparse network, while AutoGrow always keeps full-sized filter sets without any reallocation.

## 4.3 COMPARING WITH NAS METHODS

As a fair comparison with NAS methods involving search and re-training phases, we also divide our method into growing and training phases. Specifically, we grow layers and channels from the *Bottleneck4ResNet* seed architecture directly on ImageNet by setting $\lambda_1^{base} = 2.0$, $\lambda_2^{base} = 0.1$ and the parameter budget under 7M. Then we resume training the grown architecture and compare with existing NAS methods in terms of parameters, top-1 validation accuracy and V100 GPU hours required by the search or growing stages, as shown in Table 4. Note that DARTS (Liu et al., 2019) conducts search on CIFAR-10, then transfers to ImageNet instead of direct search. This is because DARTS operates on a supernet by including all the candidate paths and suffers from GPU memory explosion. In terms of epoch-wise FLOPs, results shown in Figure 1(c) are for training an equivalent of ResNet-20 on CIFAR-10 in comparison with DARTS and Provable channel pruning approach (Liebenwein et al., 2020). Also note that the EfficientNet-B0 architecture, included in Table 4, is generated by grid search in the MnasNet search space, thus having the same heavy search cost. To achieve the reported performance, EfficientNet-B0 utilizes additional squeeze-and-excitation (SE) (Hu et al., 2018) modules, AutoAugment (Cubuk et al., 2019), as well as much longer re-training epochs on ImageNet.

ProxylessNet still starts with an over-parameterized supernet, but applies a pruning-like search method by binarizing the architecture parameters and forcing only one path to be activated at search-time. This enables directly searching on ImageNet, achieving $200\times$ more search cost savings than MnasNet. Contrasting with ProxylessNet, our method progressively adds filters and layers to simple seed architectures while maintaining sparsification, which leads to savings of not only epoch-wise computation but also memory consumption, enabling faster, larger-batch training. As such, we further save $45\%$ of the GPU search hours, while achieving comparable accuracy-parameter trade-offs.

Table 4: Performance comparing with NAS methods AmoebaNet-A (Real et al., 2019), MnasNet (Tan et al., 2019), EfficientNet-B0 (Tan & Le, 2019), DARTS (Liu et al., 2019) and ProxylessNet (Cai et al., 2019) on ImageNet.

| Method | Params | Top-1 | Search/Grow Cost |
|---|---|---|---|
| AmoebaNet-A | 5.1M | 74.5% | 76K GPU hours |
| MnasNet | **4.4M** | 74.0% | 40K GPU hours |
| EfficientNet-B0 | 5.3M | **77.1**% (+SE) | 40K GPU hours |
| DARTS | 4.7M | 73.1% | N/A |
| ProxylessNet(GPU) | 7.1M | 75.1% | 200 GPU hours |
| Ours | 6.8M | 74.3% | **80** GPU hours |
| Ours | 6.7M | 74.8% | 110 GPU hours |
| Ours | 6.9M | 75.1% | 140 GPU hours |

## 4.4 ANALYSIS

**Training Cost Savings.** Figure 4 illustrates our sparsification dynamics, showing epoch-wise FLOPs while growing a ResNet-20. Appendix A.8 presents additional visualizations. Even with fully parallel GPU hardware, starting with few filters and layers in the network will ultimately save wall-clock time, as larger batch training (Goyal et al., 2017) can always be employed to fill the hardware.

Figure 5 shows validation accuracy, model complexity, and layer count while growing *Basic3ResNet*. Complexity is measured as the model parameters ratio of AutoGrow's target model. At the end of 160 epochs, our method's validation accuracy is 92.36% , which is higher than AutoGrow's 84.65% at 360 epochs. We thus require fewer fine-tuning epochs to achieve a final 94.50% accuracy on CIFAR.

**Budget-Aware Growing.** In Figure 6, for ResNet-20 on CIFAR-10, we compare architectures obtained by (1) *uniform pruning*: a naive pre-defined pruning method that prunes the same percentage of channels in each layer, (2) *ours*: variants of our method by setting different model parameter sparsities as target budgets during growing, and (3) *direct design*: our grown architectures re-initialized with random weights and re-trained.

Table 5: Comparison with random pruning baseline on CIFAR-10.

| Model | Method | Val Acc(%) | Params(M) |
|---|---|---|---|
| VGG-16 | Random | $90.01 \pm 0.69$ | $0.770 \pm 0.050$ |
| | Ours | $\mathbf{92.50 \pm 0.10}$ | $0.754 \pm 0.005$ |
| ResNet-20 | Random | $89.18 \pm 0.55$ | $0.100 \pm 0.010$ |
| | Ours | $\mathbf{90.91 \pm 0.07}$ | $0.096 \pm 0.002$ |
| WRN-28-10 | Random | $92.26 \pm 0.87$ | $3.440 \pm 0.110$ |
| | Ours | $\mathbf{95.32 \pm 0.11}$ | $3.443 \pm 0.010$ |

In most budget settings, our growing method outperforms direct design and uniform pruning, demon-

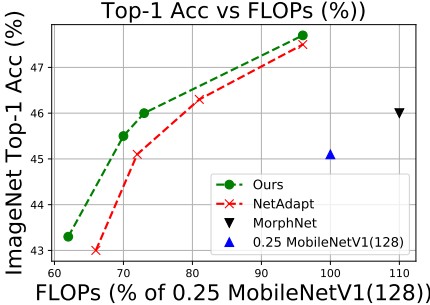

Figure 3: Performance/FLOPs trade-offs for pruned MobileNetV1 on ImageNet.

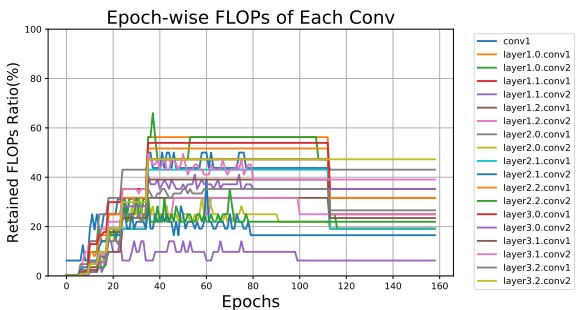

Figure 4: Epoch-wise training FLOPs for channel growing a ResNet-20.

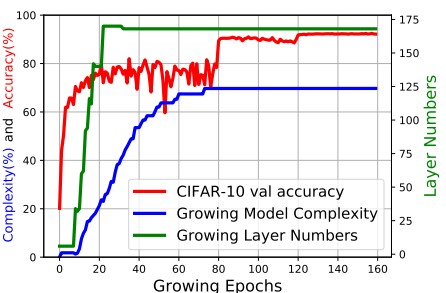

Figure 5: Tracking validation accuracy, complexity and layers for *Basic3ResNet* growing.

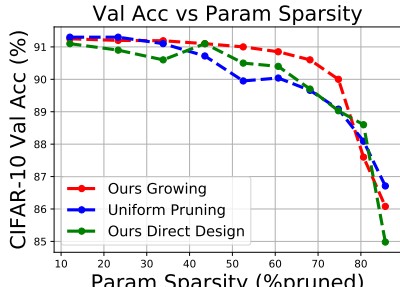

Figure 6: Pruned architectures obtained by ablated methods with different parameter sparsity.

strating higher parameter efficiency. Our method also appears to have positive effect in terms of regularization or optimization dynamics, which are lost if one attempts to directly train the final compact structure. Appendix A.9 investigates FLOPs-based budget targets.

**Comparing with Random Baseline.** In addition to the *uniform pruning* baseline in Figure 6, we also compare with a random sampling baseline to further separate the contribution of our configuration space and growing method, following the criterion in (Xie et al., 2019a; Li & Talwalkar, 2019; Yu et al., 2020; Radosavovic et al., 2019). Specifically, this random baseline replaces the procedure for sampling entries of $q$ in Eq. 6. Instead of using sampling probabilities derived from the learned mask parameters $s$, it samples with fixed probability. As shown in Table 5, our method consistently performs much better than this random baseline. These results, as well as the more sophisticated baselines in Figure 6, demonstrate the effectiveness of our growing and pruning approach.

**Temperature Scheduler.** We compare our structure-wise temperature control to a global one in channel growing experiments on CIFAR-10 using VGG-16, ResNet-20, and WideResNet-28-10. Table 1 results use our structure-wise scheduler. To achieve similar sparsity with the global scheduler, the corresponding models suffer accuracy drops of $1.4\%$, $0.6\%$, and $2.5\%$. With the global scheduler, optimization of mask variables stops early in training and the following epochs are equivalent to directly training a fixed compact network. This may force the network to be stuck with a suboptimal architecture. Appendix A.10 investigates learning rate and temperature schedule interactions.

# 5 CONCLUSION

We propose a simple yet effective method to grow efficient deep networks via structured continuous sparsification, which decreases the computational cost not only of inference but also of training. The method is simple to implement and quick to execute; it automates the network structure reallocation process under practical resource budgets. Application to widely used deep networks on a variety of tasks shows that our method consistently generates models with better accuracy-efficiency trade-offs than competing methods, while achieving considerable training cost savings.

**Acknowledgments.** This work was supported by the University of Chicago CERES Center for Unstoppable Computing and the National Science Foundation under grant CNS-1956180.

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

# A APPENDIX

## A.1 MORE DETAILED ANALYSIS FOR BUDGET-AWARE GROWING

Conducting grid search on trade-off parameters $\lambda_1$ and $\lambda_2$ is prohibitively laborious and time-consuming. For example, to grow an efficient network on CIFAR-10, one needs to repeat many times a run of 160-epochs training, and then pick the best model from all grown candidates. To avoid this tedious iterative process, instead of using constants $\lambda_1$ and $\lambda_2$, we dynamically update $\lambda_1$ and $\lambda_2$ in our one-shot budget-aware growing optimization.

Here we discuss about how budget-aware dynamic growing works in our method. Without loss of generality, we derive the $m_c$'s SGD update rule for the $\ell_0$ regularization term in Eq. 3 as:

$$m_c \leftarrow m_c - \eta \lambda_1^{base} \Delta u \frac{\delta \ell}{\delta m_c} - \eta \mu \lambda_1^{base} \Delta u m_c \tag{8}$$

where $\eta$ is the learning rate and $\mu$ is the weight decay factor. At the beginning of growing epochs, when the architecture is very over-sparsified, $\Delta u$ and $\lambda_1^{base} \Delta u$ are negative values. Then $m_c$'s update is along the **opposite** direction of the $\ell_0$ regularization term's gradients, encouraging $m_c$'s sparsification. As a result, some zero-valued $m_c$ will be activated and the model complexity is strongly increased to acquire enough capacity for successful training. Then, growing becomes gradually weaker as the network's sparsity approaches the budget ($\Delta u$ to zero). Note that if the architecture is over-parameterized, $\Delta u$ and $\lambda_1^{base} \Delta u$ become positive and SGD's update rule is the same as that of $\ell_0$ regularization. As such, our budget-aware growing can automatically and dynamically adapt the architecture complexity not only based on the task loss $L_E$ but also on the practical budget requirements in the one-shot training process.

We also note that NAS methods usually use the validation accuracy as a target during their architecture optimization phase, which may require some prior knowledge of validation accuracy on a given dataset. Our growing procedure chooses sparsity budget instead of accuracy as target because: (1) During growing, validation accuracy is influenced not only by architectures but also model weights. Directly using $\Delta acc$ may lead to sub-optimal architecture optimization. (2) A sparsity budget target is more practical and easier to set according to target devices for deployment.

## A.2 DETAILS OF EVALUATION DATASETS

Evaluation is conducted on various tasks to demonstrate the effectiveness of our proposed method. For image classification, we use CIFAR-10 (Krizhevsky et al., 2014) and ImageNet (Deng et al., 2009): CIFAR-10 consists of 60,000 images of 10 classes, with 6,000 images per class. The train and test sets contain 50,000 and 10,000 images respectively. ImageNet is a large dataset for visual recognition which contains over 1.2M images in the training set and 50K images in the validation set covering 1,000 categories. For semantic segmentation, we use the PASCAL VOC 2012 (Everingham et al., 2015) benchmark which contains 20 foreground object classes and one background class. The original dataset contains 1,464 (train), 1,449 (val), and 1,456 (test) pixel-level labeled images for training, validation, and testing, respectively. The dataset is augmented by the extra annotations provided by (Hariharan et al., 2011), resulting in 10,582 training images. For language modeling, we use the word level Penn Treebank (PTB) dataset (Marcus et al., 1993) which consists of 929k training words, 73k validation words, and 82k test words, with 10,000 unique words in its vocabulary.

## A.3 UNPRUNED BASELINE MODELS

For CIFAR-10, we use VGG-16 (Simonyan & Zisserman, 2015) with BatchNorm (Ioffe & Szegedy, 2015), ResNet-20 (He et al., 2016) and WideResNet-28-10 (Zagoruyko & Komodakis, 2016) as baselines. We adopt a standard data augmentation scheme (shifting/mirroring) following (Lin et al., 2013; Huang et al., 2016), and normalize the input data with channel means and standard deviations. Note that we use the CIFAR version of ResNet-20[1], VGG-16[2], and WideResNet-28-10[3]. VGG-16, ResNet-20, and WideResNet-28-10 are trained for 160, 160, and 200 epochs, respectively, with a

---

[1]https://github.com/akamaster/pytorch_resnet_cifar10/blob/master/resnet.py

[2]https://github.com/kuangliu/pytorch-cifar/blob/master/models/vgg.py

[3]https://github.com/meliketoy/wide-resnet.pytorch/blob/master/networks/wide_resnet.py

batch size of 128 and initial learning rate of 0.1. For VGG-16 and ResNet-20, we divide learning rate by 10 at epochs 80 and 120, and set the weight decay and momentum as $10^{-4}$ and 0.9. For WideResNet-28-10, the learning rate is divided by 5 at epochs 60, 120, and 160; the weight decay and momentum are set to $5 \times 10^{-4}$ and 0.9. For ImageNet, we train the baseline ResNet-50 and MobileNetV1 models following the respective papers. We adopt the same data augmentation scheme as in (Gross & Wilber, 2016) and report top-1 validation accuracy. For semantic segmentation, the performance is measured in terms of pixel intersection-over-union (IOU) averaged across the 21 classes (mIOU). We use Deeplab-v3-ResNet-101[4] (Chen et al., 2017) as the baseline model following the training details in (Chen et al., 2017). For language modeling, we use vanilla two-layer stacked LSTM (Zaremba et al., 2014) as a baseline. The dropout keep ratio is 0.35 for the baseline model. The vocabulary size, embedding size, and hidden size of the stacked LSTMs are set as 10,000, 1,500, and 1,500, respectively, which is consistent with the settings in (Zaremba et al., 2014).

## A.4    MOBILENETV1 CHANNEL GROWING ON IMAGENET

To further validate the effectiveness of the proposed method on compact networks, we grow the filters of MobileNetV1 on ImageNet and compare the performance of our method to the results reported directly in the respective papers, as shown in Table 6. In MobileNetV1 experiments, following the same setting with Netadapt (Yang et al., 2018), we apply our method on both (1) small setting: growing MobileNetV1(128) with 0.5 multiplier while setting the original model's multiplier as 0.25 for comparison and (2) large setting: growing standard MobileNetV1(224) while setting the original model's multiplier as 0.75 for comparison. Note that MobileNetV1 is one of the most compact networks, and thus is more challenging to simplify than other larger networks. Our lower-cost growing method can still generate a sparser MobileNetV1 model compared with competing methods.

Table 6: Overview of the pruning performance of each algorithm on MobileNetV1 ImageNet.

| Model | Method | Top-1 Val Acc(%) | FLOPs(%) | Train-Cost Savings(×) |
|---|---|---|---|---|
| MobileNet V1(128) | Original(25%) | 45.1 (+0.0) | 100 | 1.0(×) |
| | MorphNet | 46.0 (+0.9) | 110 | 0.9(×) |
| | Netadapt | **46.3 (+1.2)** | 81 | 1.1(×) |
| | Ours | 46.0 (+0.9) | **73** | **1.7**(×) |
| MobileNet V1(224) | Original(75%) | 68.8 (+0.0) | 100 | 1.0(×) |
| | Netadapt | 69.1 (+0.3) | 87 | 1.2(×) |
| | Ours | **69.3 (+0.5)** | **83** | **1.5**(×) |

## A.5    DEEPLAB-V3-RESNET-101 ON PASCAL VOC 2012

We also test the effectiveness of our proposed method on a semantic segmentation task by growing a Deeplab-v3-ResNet-101 model's filter numbers from scratch directly on the PASCAL VOC 2012 dataset. We apply our method to both the ResNet-101 backbone and ASPP module. Compared to the baseline, the final generated network reduces the FLOPs by 58.5% and the parameter count by 49.8%, while approximately maintaining mIoU (76.5% to 76.4%). See Table 7.

Table 7: Results on the PASCAL VOC dataset.

| Model | Method | mIOU | Params(M) | FLOPs(%) | Train-Cost Savings(×) |
|---|---|---|---|---|---|
| Deeplab -v3- ResNet101 | Original | 76.5 (-0.0) | 58.0 (100%) | 100 | 1.0(×) |
| | L1-Pruning | 75.1 (-1.4) | 45.7 (78.8%) | 62.5 | 1.3(×) |
| | Ours | **76.4 (-0.1)** | **29.1 (50.2%)** | **41.5** | **2.3**(×) |

---

[4]https://github.com/chenxi116/DeepLabv3.pytorch

A.6   EXTENSION TO RECURRENT CELLS ON PTB DATASET

We focus on LSTMs (Hochreiter & Schmidhuber, 1997) with $l_h$ hidden neurons, a common variant[5] of RNNs that learns long-term dependencies:

$$
\begin{aligned}
f_t &= \sigma_g((W_f \odot (\mathbf{e}m_c^T))x_t + (U_f \odot (m_c m_c^T))h_{t-1} + b_f) \\
i_t &= \sigma_g((W_i \odot (\mathbf{e}m_c^T))x_t + (U_i \odot (m_c m_c^T))h_{t-1} + b_i) \\
o_t &= \sigma_g((W_o \odot (\mathbf{e}m_c^T))x_t + (U_o \odot (m_c m_c^T))h_{t-1} + b_o) \\
\tilde{c}_t &= \sigma_h((W_c \odot (\mathbf{e}m_c^T))x_t + (U_c \odot (m_c m_c^T))h_{t-1} + b_c) \\
c_t = f_t \odot c_{t-1} + i_t \odot \tilde{c}_t, &\quad h_t = o_t \odot \sigma_h(c_t) \quad s.t. \quad m_c \in \{0,1\}^{l_h}, \mathbf{e} = 1^{l_h},
\end{aligned}
\tag{9}
$$

where $\sigma_g$ is the sigmoid function, $\odot$ denotes element-wise multiplication and $\sigma_h$ is the hyperbolic tangent function. $x_t$ denotes the input vector at the time-step $t$, $h_t$ denotes the current hidden state, and $c_t$ denotes the long-term memory cell state. $W_f, W_i, W_o, W_c$ denote the input-to-hidden weight matrices and $U_f, U_i, U_o, U_c$ denote the hidden-to-hidden weight matrices. $m_c$ is binary indicator and shared across all the gates to control the sparsity of hidden neurons.

We compare our proposed method with ISS based on vanilla two-layer stacked LSTM. As shown in Table 8, our method finds more compact model structure at lower training cost, while achieving similar perplexity on both validation and test sets. These improvements may be due to the fact that our method dynamically grows and prunes the hidden neurons from very simple status towards a better trade-off between model complexity and performance than that of ISS, which simply uses the group lasso to penalize the norms of all groups collectively for compactness.

Table 8: Results on the PTB dataset.

| Method | Perplexity (val,test) | Final Structure | Weight(M) | FLOPs(%) | Train-Cost Savings($\times$) |
|---|---|---|---|---|---|
| Original | (82.57, 78.57) | (1500, 1500) | 66.0M (100%) | 100 | 1.0($\times$) |
| ISS | (82.59, **78.65**) | (373, 315) | 21.8M (33.1%) | 13.4 | 3.8($\times$) |
| Ours | (**82.46**, 78.68) | (**310, 275**) | **20.6M (31.2%)** | **11.9** | **5.1($\times$)** |

A.7   VARIANTS OF INITIAL SEED ARCHITECTURE

In Table 9, we make a detailed comparison among initial seed architecture variants of ours and AutoGrow (Wen et al., 2020). For both ours and AutoGrow, "Basic" and "Bottleneck" refer to ResNets with standard basic and bottleneck residual blocks, while "PlainLayers" refers to stacked convolutional, batch normalization, and ReLU layer combinations. Similar with standard ResNets, for variants of the seed architecture, we adopt three stages for CIFAR-10 and four stages for ImageNet. PlainNets can be obtained by simply removing shortcuts from these ResNet seed variants with equal stage numbers. For each stage, we start from only one growing unit, within which initial filter numbers are also initialized at one for channel growing.

A.8   TRACK OF ANY-TIME SPARSIFICATION DURING CHANNEL GROWING

Figure 7 and Figure 8 show the dynamics of train-time growing channel ratios of ResNet-20 and VGG-16 on CIFAR-10, respectively. To better analyze the growing patterns, we visualize the channel dynamics grouped by stages in Figure 9 for ResNet-20 and Figure 10 for VGG-16, respectively. Note that, for VGG-16, we divide it into 5 stages based on the pooling layer positions and normalize channel ratios by 0.5 for better visualization. We see that our method grows more channels of earlier layers within each stage of ResNet-20. Also, the final channel sparsity of ResNet-20 is more uniform due to the residual connections.

Table 9: A detailed comparison among seed architecture variants of our method and AutoGrow (Wen et al., 2020). In growing units term, "Basic" and "Bottleneck" refer to ResNets with standard basic and bottleneck residual blocks while "PlainLayers" refers to standard convolutional layer, BN, and ReLu layer combinations in VGG-like networks without shortcuts.

| Families | Variants | Methods | Channel Growing | Growing Units | Stages | Shortcuts |
|---|---|---|---|---|---|---|
| ResNet | *Basic3ResNet* | Ours | ✓ | Basic | 3 | ✓ |
| | | AutoGrow | ✗ | | | ✓ |
| | *Bottleneck4ResNet* | Ours | ✓ | Bottleneck | 4 | ✓ |
| | | AutoGrow | ✗ | | | ✓ |
| VGG-like | *Plain3Net* | Ours | ✓ | PlainLayers | 3 | ✗ |
| | | AutoGrow | ✗ | | | ✗ |
| | *Plain4Net* | Ours | ✓ | PlainLayers | 4 | ✗ |
| | | AutoGrow | ✗ | | | ✗ |

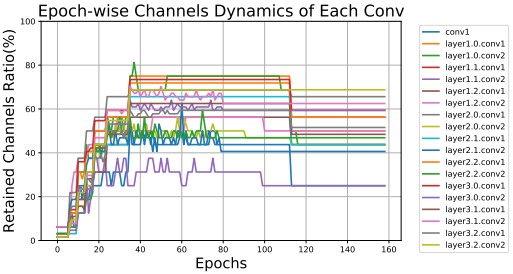

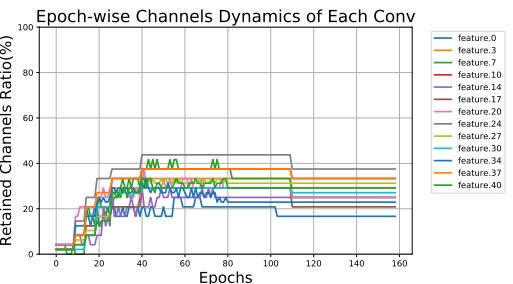

Figure 7: Epoch-wise retained channel ratio dynamics for each layer in ResNet-20.

Figure 8: Epoch-wise retained channel ratio dynamics for each layer in VGG-16.

### A.9 FLOPS-BASED BUDGET-AWARE GROWING

We also investigate the effectiveness of setting a FLOPs target for budget-aware growing in Figure 11. We observe similar trends among *uniform pruning*, *ours growing*, and *ours direct design*: in most FLOPs budget settings, our growing method outperforms direct design and uniform pruning. We also observe that when setting extreme sparse FLOPs target (*e.g.,* 85%), our method achieves lower accuracy than the other two variants. The reason is that our channel growing is forced to only grow architectures from ∼ 99% sparsity up to ∼ 85% FLOPs and ∼ 90% parameters sparsity, during which models cannot acquire enough capacity to be well trained.

### A.10 INTERACTIONS BETWEEN LEARNING RATE AND TEMPERATURE SCHEDULERS

Two factors influence the growing optimization speed in our method: temperature and learning rate, which are hyperparameters controlled by their respective schedulers. We first visualize the structure-wise separate temperature dynamics in Figure 12 by averaging temperatures per layer during ResNet-20 channel growing on CIFAR-10. We see that temperatures are growing with different rates for channels. Usually, low learning rate and high temperature in late training epochs make the network growing optimization become very stable. In Figure 13, we deliberately decay $\gamma$ in the temperature scheduler, mirroring the learning rate decay schedule, in order to force growing until the end. As shown in Figure 14, our method is still adapting some layers even at the last epoch. We find that such instability degrades performance, since some newly grown filters may not have enough time to become well trained.

---

[5]The proposed configuration space can be readily applied to the compression of GRUs (Cho et al., 2014) and vanilla RNNs.

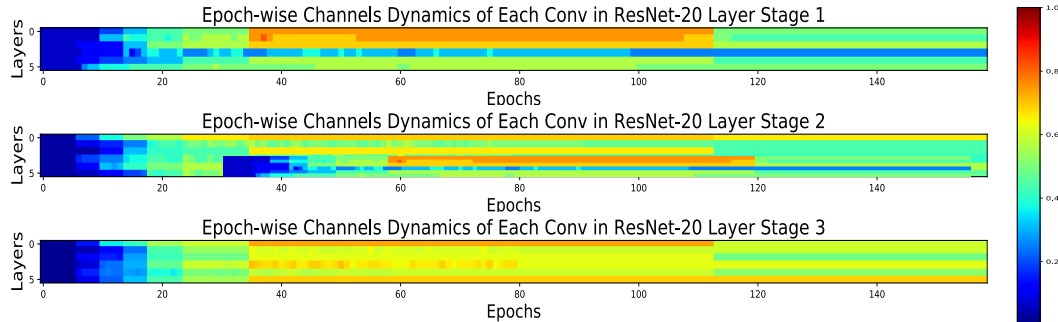

Figure 9: Visualization of retained channel ratio dynamics for each stage in ResNet-20.

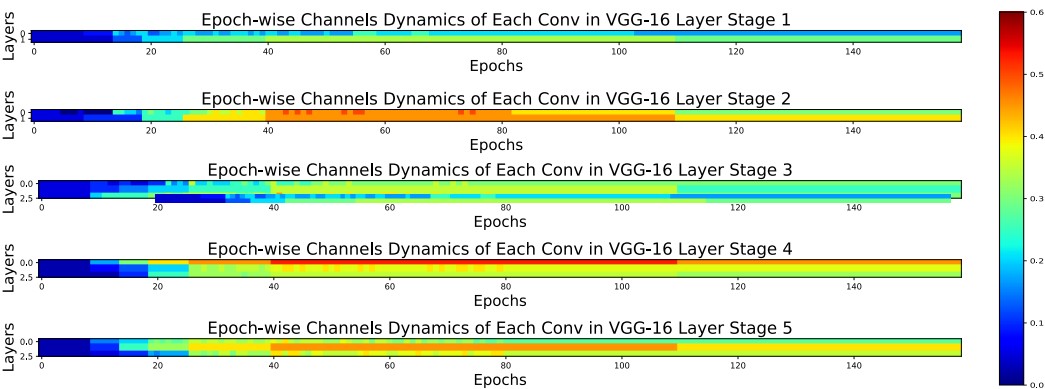

Figure 10: Visualization of retained channel ratio dynamics for each stage in VGG-16.

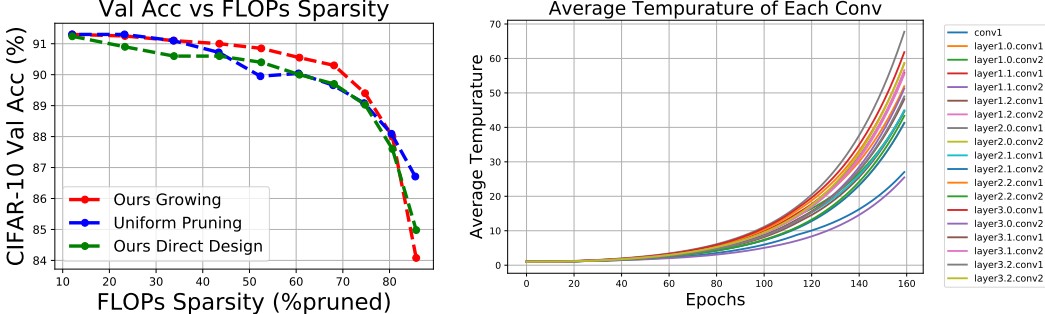

Figure 11: Pruned architectures obtained by ablated methods with different FLOPs sparsity.

Figure 12: Structure-wise separate temperature dynamics in channel growing.

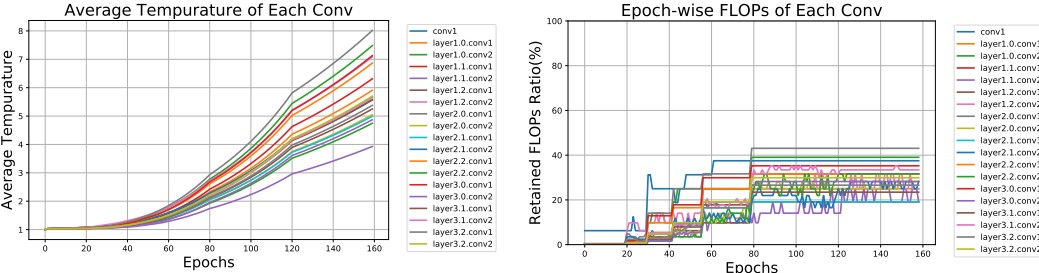

Figure 13: Structure-wise separate decayed temperature dynamics in channel growing.

Figure 14: Track of epoch-wise train-time FLOPs for channel growing in ResNet-20.

