# OpenReview forum: "Growing Efficient Deep Networks by Structured Continuous Sparsification"
_ICLR.cc/2021/Conference — ICLR 2021 Oral_

### Official Review · AnonReviewer2 · 2020-10-28
**Reviews**

**Rating:** 7
**Confidence:** 4

**Review:**

Summary:

  This paper proposes a NAS-type work for growing a small network to a large network by adding channels and layers gradually. The authors apply the method to both CNN and LSTM networks.

Strong points:

  1. This paper is well-written and shows good results.

  2. The proposed algorithm is sound and effective. E.g. use less wall-time as compared to other NAS approaches.


Weak points:

  1. It seems that the number of channels and number of layers still need to be predefined (the mask size).


Questions:


  1. How does the FLOPs reduction translate to runtime saving?

  2. What is the target sparsity u in the experiments?

  3. When growing with layers, does the author observe any middle layer is dropped and then recovered? If so, does it happen frequently?

  4. At section 4.2 and 4.3, what is the size of the channel/layer mask? I believe the author still needs to define the upper bound of the network can grow. If so, does the upper bound affect the optimization?  Or the proposed method gradually expands the mask?

  5. In table 4, I think Efficient-B0 should be taken into consideration as it is a recent representative approach.

After rebuttal:

The authors' rebuttal addressed all my questions and I upgrade my rating.

---

> ### Author Response · Authors · 2020-11-24
> **response to AnonReviewer2**
>
> Thank you for the review and comments. We address your points individually below.
>
> **Q: How does the FLOPs reduction translate to runtime savings?**
>
> **A:** Our method performs structured sparsification at two levels, removing entire layers or filters (channels) within a layer. This directly translates into runtime savings as the resulting networks have reduced depth and width. Shallower networks have correspondingly lower latency. Reduced width means less computation is required for a layer; even on a highly parallel GPU, this translates into energy savings or speedup via use of larger batch sizes. Structured sparsification notably differs from methods which prune individual weights. Unlike our method, the latter may depend upon hardware efficiently supporting sparse matrix operations in order to realize savings.
>
> **Q: What is the target sparsity u in the experiments?**
>
> **A:** In Section 4.1, we set *u* as 0.95 (VGG-16), 0.65 (ResNet-20), 0.90 (WRN-28-10), 0.4 (ResNet-50), 0.6 (MobileNetV1), 0.5 (Deeplab-V3-ResNet101) and 0.6 (2-stacked-LSTM) to compare with the channel pruning methods. In Section 4.2 and 4.3, we set *u* to 0.5 when comparing with AutoGrow and NAS methods.
>
> **Q: any middle layer is dropped and then recovered?**
>
> **A:** Yes, we have observed middle layers dropped and then recovered. Here are some statistics.
>
> We calculate frequency by simply counting: how many layers in the final architecture undergo a drop then recovery, i.e., the associated indicators flip to 0 then back to 1.
>
> In the CIFAR-10 Basic3ResNet growing experiments in Table 3, the final layer architecture is 23-29-31, we observe that "dropped and then recovered" phenomena happen in 4-7-5 of final architecture's layers. The overall frequency is (4+7+5)/(23+29+31) = 19%.
>
> In the ImageNet Bottleneck4ResNet growing experiments, the frequency is (0+2+1+2)/(5+6+5+7) = 22%.
>
> **Q: size of channel/layer mask, predefined upper bound on number of channels and number of layers?**
>
> **A:** Our method does not require setting any upper bound on the number of channels or layers.
>
> In experiments, we did predefine a maximum mask size to simplify implementation. This was sufficient to allow comparison with other methods operating on similar target model scales. Here are the upper bounds on mask size, per stage of the network, used in Sec. 4.2 and 4.3:
>
> | Property          | Sec. 4.2 (CIFAR) | Sec. 4.2 (ImageNet) | Sec. 4.3       |
> | :---------------  | :----------------| :------------------ | :------------- |
> | Channel Mask Size | 16,32,64         | 64,128,256,512      | 64,128,256,256 |
> | Layer Mask Size   | 40,40,40         | 8,8,8,8             | 4,4,4,4        |
>
> A straightforward extension of our current implementation would remove any limits on channels or layers, while minimally impacting efficiency: reallocate and expand the appropriate mask and parameter arrays when all of their elements are in use. This is precisely the same as a classic variable-length array implementation that dynamically grows or shrinks via an occasional reallocation and copy procedure.
>
> **Q: Table 4, Efficient-B0**
>
> **A:** Thanks for the suggestion. We have added Efficient-B0 [1] to Table 4. Note that the Efficient-B0 architecture is generated by grid search in the MnasNet [2] search space, implying the search cost is in the same large-scale range as MnasNet, i.e., 40,000 GPU hours.
>
> [1] Mingxing Tan and Quoc V. Le. "EfficientNet: Rethinking Model Scaling for Convolutional Neural Networks." ICML 2019.
>
> [2] Mingxing Tan, Bo Chen, Ruoming Pang, Vijay Vasudevan, and Quoc V Le. "MnaNnet: Platform-aware neural architecture search for mobile." CVPR 2019.

---

### Official Review · AnonReviewer1 · 2020-10-28
**Big idea, with extensive experiments**

**Rating:** 7
**Confidence:** 4

**Review:**

**Pros**:

1. This paper nicely unifies two different classes of approaches (NAS + sparsity) for determining the topology of neural networks. They are combined into a single optimization problem, with binary indicators on network components and connections.

2. Experiments illustrate the behavior of the method. It is good to see that the experiments dig a big deeper than end-result accuracy. For instance, the "budget-aware growing" is shown well to work as described by Fig 3.

**Cons**:

3. No attention to random seeding.

The sparsification dynamics seem likely to change somewhat from one run to the next. The submission does not describe how random seeding was done for training. Multiple runs with different seeds are not shown, and the distribution of accuracies across runs is unknown. Attention to randomness for this kind of training process seems especially important given the variances in results in the Lottery Ticket hypothesis paper.

4. No comparison to simple random baseline.

A large portion of the method consists of a search method over the space of possible sparse networks, combining it with growing the network to get a NAS-like method. It has been observed, though, that in a sufficiently general space of this kind one can randomly sample connections and see high accuracies. So to identify the sources of empirical gains, it is good to consider experimental baselines, such as random sampling, that separate the contribution of the search space and the search method:

  * Xie et. al. "Exploring randomly wired neural networks for image recognition"
  * Li & Talkwaker "Random search and reproducibility for neural architecture search"
  * Yu et. al. "Evaluating the Search Phase of Neural Architecture Search"
  * Radosavovic et. al. "On Network Design Spaces for Visual Recognition."

Note that the submission's method also is randomly choosing connections, through a somewhat involved process that also accounts for the observed sparsity of G during training. The simplest baseline seems to be the "uniform pruning" described in Section 4.4. This only ablates part of the method that doesn't seem to meet the same criterion here.

5. Incomplete illustration of the cost/accuracy tradeoff.

The gold standard for comparison in both sparse-neural-network papers and NAS is to consider the accuracy at a range of different model costs. See for example:

  * Blalock et. al. "What is the state of neural network pruning?" Figs 1, 3
  * (Yang et. al., 2018) Figs 5-9

This clearly illustrates whether a method is overall better (i.e. produces better models across the entire pareto frontier), or is only better for some ranges or on one metric. For a result like the first one in Table 5 in the submission, it is unclear which model is better: they may simply be considering different points on the same cost/accuracy curve.

6. As a less important aside, "budget-aware growing" seems to be an ad-hoc reinvention of something similar to an Augmented Lagrangian method. Explicitly describing the differences from standard optimization techniques might be good.

**Reasoning for rating**:

While the experiments are extensive, I think they miss the key comparisons that show how useful the method and each of its components is. Given that many different innovations are included in the submission, it may be a muddle for follow-up research to sort out how good each individual one is.

**Misc comments**

  * Check spacing around (6) in Algorithm 1
  * Colon instead of comma after "trainable variables" in §4.1
  * "For better analyze the growing patterns" -> "To better analyze the growing patterns" on page 14
  * Wortsman et. al. "Discovering Neural Wirings" is another closely related work at the intersection of NAS and pruning. (with major differences from the submission)

**After rebuttal**

The authors have gone above and beyond in providing additional experimental results. All of the points raised above that deal with methodological issues are completely addressed.

The sole significant weakness that remains is the lack of the kind of ablation/component studies that would justify individual design decisions. I do not disagree with the authors that this will be difficult for this work, but I still feel they would have been helpful for researchers who will be building upon this method.

---

> ### Author Response · Authors · 2020-11-24
> **response to AnonReviewer1**
>
> Thank you for the review and comments. We address your points individually below.
>
> **Q: random seeding**
>
> **A:** We have re-run our pruning method on CIFAR-10 (matching original experiments in Table 1) with 5 different random seeds and report results as follows, written as mean (+/- standard deviation):
>
> | model     | CIFAR Acc(%)      | Params(M)        | FLOPs(%)        | Train-cost savings(x) |
> | :-------  | :---------------- | :--------------- | :-------------- | :------------- |
> | VGG-16    | 92.50 (+/-0.10)   | 0.754 (+/-0.005) | 13.55 (+/-0.03) | 4.95 (+/-0.17) |
> | ResNet-20 | 90.91 (+/-0.07)   | 0.096 (+/-0.002) | 50.20 (+/-0.01) | 2.40 (+/-0.09) |
> | WRN-28-10 | 95.32 (+/-0.11)   | 3.443 (+/-0.010) | 28.25 (+/-0.04) | 3.12 (+/-0.11) |
>
> These results demonstrate high consistency across random seeds. They indicate that the advantages we originally observed in comparison to other methods, in terms of accuracy improvement, parameter and/or FLOP reduction, and training cost savings, are beyond the range explainable by chance.
>
> To the final version of the paper, we will include these variance statistics, as well as those for re-running from different random initialization in all competing methods, which requires additional time to complete.
>
> **Q: comparison to simple random baseline**
>
> **A:** Section 4.4 includes comparison to a "uniform pruning" baseline; we expected this would be the strongest simple baseline against which to compare. We agree with the suggestion of also comparing to a simple random baseline and have conducted this experiment. This new random baseline replaces the procedure for sampling values of *q* in Equation 6. Instead of using sampling probabilities derived from the learned parameters *s*, it samples with fixed probability. On CIFAR-10 (matching the setting in Table 1), we obtain the following results, averaged across 5 runs:
>
> | Model     | Method | Val Acc(\%)     | Params(M)        |
> | :-------- | :----- | :-------------- | :--------------- |
> | VGG-16    | random | 90.01 (+/-0.69) | 0.770 (+/-0.050) |
> | VGG-16    | ours   | 92.50 (+/-0.10) | 0.754 (+/-0.005) |
> | ResNet-20 | random | 89.18 (+/-0.55) | 0.100 (+/-0.010) |
> | ResNet-20 | ours   | 90.91 (+/-0.07) | 0.096 (+/-0.002) |
> | WRN-28-10 | random | 92.26 (+/-0.87) | 3.440 (+/-0.110) |
> | WRN-28-10 | ours   | 95.32 (+/-0.11) | 3.443 (+/-0.010) |
>
> For each model, our method produces trained networks with both substantially higher validation accuracy and comparable or fewer parameters than the random baseline. We have updated Section 4.4 to include these results and corresponding discussion.
>
> **Q: incomplete illustration of the cost/accuracy tradeoff**
>
> **A:** We have conducted additional experiments to illustrate the cost/accuracy tradeoff curve.
>
> To address the concern about first result in the original Table 5 (MobileNetV1 on ImageNet), for both NetAdapt and our method, we train four model variants of different size, and plot accuracy vs FLOPs trade-offs in Figure 3 of our revised paper. Our method dominates NetAdapt, achieving higher accuracy using fewer FLOPs.
>
> We have also updated Table 4 with two additional models trained by our method. Using a longer training time, but still faster than ProxylessNet, our method produces a network that matches ProxylessNet in accuracy while having fewer parameters:
>
> | Method            | Params(M) | Top-1 Acc(%) | Search/Grow Cost |
> | :---------------- | :-------- | :----------- | :--------------- |
> | ProxylessNet(GPU) | 7.1       | 75.1         | 200 GPU hours    |
> | Ours-1            | 6.8       | 74.3         |  80 GPU hours    |
> | Ours-2            | 6.7       | 74.8         | 110 GPU hours    |
> | Ours-3            | 6.9       | 75.1         | 140 GPU hours    |
>
> **Q: budget-aware growing and Augmented Lagrangian method**
>
> **A:** It is not our intention to claim any novelty in terms of optimization techniques. Indeed, the budget-aware growing procedure in Appendix A.1 does share a similar spirit to an Augmented Lagrangian method, in that it periodically revises an unconstrained objective function in order to drive the system towards a (budget) constraint.
>
> **Q: many different innovations**
>
> **A:** Though our method has several components, it is not a collection of orthogonal innovations. Rather, these components are designed and woven together to implement a high-level vision: efficiently train deep networks by growing them over time while pruning away useless structure. Continuous sparsification, temperature schedules, and sampling form the technical basis which enables the simultaneous growing and pruning dynamics.
>
> **Q: misc**
>
> **A:** We have revised accordingly and added citation of Wortsman et. al.

---

### Official Review · AnonReviewer3 · 2020-10-28
**Official Blind Review #3**

**Rating:** 7
**Confidence:** 3

**Review:**

This paper proposes a novel NAS method that searches the model architectures by grows the networks. This searching strategy determines the channel and layer configurations by assigning a binary learnable parameter m for each channel or layer. The objective is to optimize a trade-off between the model performance on the given task and the regularization on the binary indicator m.

Pros:
1. The general idea of searching the architectures by growing the networks sounds very interesting. The authors propose a novel framework to achieve their idea, and also apply some tricks to speed up and simplify the optimization (e.g. budget-aware growing and learning by continuation).
2. The paper is well-written and easy to follow.
3. The authors conduct a series of solid experiments to verify the effectiveness of their proposed methods. The experiments show the performance of channel pruning, the remarkable improvement on AutoGrow model, and the comparsion with other NAS methods.

Cons:
1. Compared to ProxylessNet, the proposed model can reduce half of the training time but does harm to the model performance.

Questions:
1. What's the exact meaning of "Top-1 valiadation accuracy"? What's the different with Top-1 accuracy? Is this metric evaluated on the valiadation set?

---

> ### Author Response · Authors · 2020-11-24
> **response to AnonReviewer3**
>
> Thank you for the review and comments. We address your points individually below.
>
> **Q: Compared to ProxylessNet, the proposed model can reduce half of the training time but does harm to the model performance.**
>
> **A:** To provide a comparison at equal accuracy, we have updated Table 4 with two additional models trained by our method. Using a longer training time, but still faster than ProxylessNet, our method produces a network that matches ProxylessNet in accuracy while having fewer parameters. Here are the results on ImageNet (also included in our reply to AnonReviewer1):
>
> | Method            | Params(M) | Top-1 Acc(%) | Search/Grow Cost |
> | :---------------- | :-------- | :----------- | :--------------- |
> | ProxylessNet(GPU) | 7.1       | 75.1         | 200 GPU hours    |
> | Ours-1            | 6.8       | 74.3         |  80 GPU hours    |
> | Ours-2            | 6.7       | 74.8         | 110 GPU hours    |
> | Ours-3            | 6.9       | 75.1         | 140 GPU hours    |
>
>
> Our method saves 30% of the training time compared to ProxylessNet (140 vs 200 GPU hours), while producing a slightly smaller network (6.9M vs 7.1M params) that has the same accuracy (75.1%).
>
> **Q: Top-1 validation accuracy**
>
> **A:** Top-1 accuracy and Top-1 validation accuracy are synonymous in our usage. Top-1 validation accuracy is exactly the Top-1 accuracy achieved by the network on the ImageNet validation set.

---

### Official Review · AnonReviewer4 · 2020-10-29
**Principled approach to growing deep networks**

**Rating:** 8
**Confidence:** 4

**Review:**

This paper proposes a new principled approach to growing deep network architectures based on continuous relaxation of discrete structure optimization combined with a sparse subnetwork sampling scheme. It starts from a simple seed architecture and dynamically grows/prunes both the layers and filters during training. Through extensive experiments, the authors show that this method produces more efficient networks while reducing the computational cost of training, still maintaining good validation accuracy, compared to other NAS or pruning/growing methods.

Strength:
(+) The proposed idea of formulating the problem as a continuous relaxation of discrete structure optimization is interesting. It seems to be a more principled approach than previous NAS or separate pruning/growing approaches.
(+) Extensive experimental results are provided to verify the superiority over recent other methods and also to show the performance behavior of the proposed method. The overall experimental setup is systematic and comprehensive.
The experiments were done on widely used deep networks on various tasks.

I only have concerns about the clarity of the notation and the representation of the figures. Specific examples are as follows:

In Eq. (3), it is said that f is the operation in Eq. (1). However, I couldn’t find f in Eq. (1).

It should be clarified how many temperature parameters β are in the proposed model. Only one or as many as channels and layers? If it is only one, it does not seem reasonable that all the probabilities growing or pruning channels and layers are the same.  Equation (7) seems to imply β to be a vector, but earlier notations (e.g. in Algo 1, Equation 6, etc.) seem to present it as a scalar.

Overall, there are confusing symbols, whether it’s a scalar or a vector. I recommend the channel and layer indicators are denoted as vectors. It seems that each channel and each layer has its unique indicator, respectively. Also, notations should include channel and layer index if they are different depending on channels and layers.

Additionally, all the experimental results shown in the main manuscript are on convolutional neural networks while the abstract mentions recurrent neural networks. The appendix has some, but very little has the main manuscript. If it’s an important part of this manuscript, the authors should include at least a brief summary of the results.

Some figures (and the text inside) are too small while containing many details, probably because of the space limit. For example, Figure 3 has many lines that are hard to analyze and texts that are not readable.

In Table 1, what’s the meaning of the underlines? I guess the second best results, but for RestNet-20, the method with the second-best FLOPs is SoftNet, not Provable. And the explanation about the boldface and underlines should be included.

---

> ### Author Response · Authors · 2020-11-24
> **response to AnonReviewer4**
>
> Thank you for the review and comments. We address your points individually below.
>
> **Q: In Eq. (3), it is said that f is the operation in Eq. (1). However, I couldn’t find f in Eq. (1).**
>
> **A:** $f$ in Eq. (3) denotes the 'conv' operation in Eq. (1). To clarify, we have replaced 'conv' in Eq. (1) with $f$.
>
> **Q: clarify how many temperature parameters $\beta$ are in the proposed model; recommend channel and layer indicators are denoted as vectors.**
>
> **A:** The total count of temperature parameters is the sum of the total number of channels in the network and the total number of layers. That is, each channel (filter) has an associated temperature parameter, as does each layer. This is identical to the situation with mask variables: per channel and per layer indicators allow for structured pruning of those components. The associated temperature depends upon the age of the corresponding component in the network, as defined by Eq. (7).
>
> Eq. (7) and the "improved temperature scheduler" subsection introduce this per-component definition of temperature. In Eq. (6), which is presented prior to this discussion, we denote temperature as a scalar, following the conventions in existing pruning or NAS work, where all components share a single temperature parameter.
>
> We have revised the paper to use bold font for all vector-valued variables.
>
> **Q: The authors should include at least a brief summary of the recurrent neural networks results.**
>
> **A:** We have added a brief summary of our LSTM experiments to Sec 4.1.
>
> **Q: Some figures (and the text inside) are too small while containing many details.**
>
> **A:** We have enlarged Figures 3, 4, and 5 in the updated version of the paper.
>
> **Q: In Table 1, what’s the meaning of the underlines? ...the method with the second-best FLOPs is SoftNet, not Provable.**
>
> **A:** Underlines denote the second best results, while bold denotes the best. In the updated paper, we have added this explanation as well as correctly denoted SoftNet as having the second-best FLOPs for ResNet-20 in Table 1.

---

### Author Response · Authors · 2020-11-24
**Paper Revision**

We thank all the reviewers, and answer questions in individual responses to each review below. We have revised the paper, incorporating suggested changes, including:

* For experiments in Table 1 (CIFAR), we now report our method's performance in the fashion of mean +/- standard deviation across 5 runs with different random seeds. In the final version, we will a include similar update for all competing methods.

* We conducted additional experiments on ImageNet to analyze the full accuracy vs FLOPs trade-off curve of our method in comparison to NetAdapt. A newly added Figure 3 shows that our method's trade-off curve dominates that of NetAdapt.

* To the ImageNet results in Table 4, we added EfficientNet-B0. In order to facilitate comparison with ProxylessNet at equal accuracy, we also trained two additional models using our method. We match ProxylessNet's accuracy, while still training faster and producing a smaller model.

* We add a random sampling baseline experiment to Section 4.4, with results displayed in a new Table 5. Our method's learned strategy significantly outperforms this random baseline.

* The main text now contains a brief summary of our LSTM experiments, referring to the Appendix for full details.

---

### Decision · Program_Chairs · 2021-01-07
**Final Decision**

**Decision:**

Accept (Oral)

**Comment:**

The paper proposes a method to grow deep network architectures over the course of training. The work has been extremely well received and has clear novelty and solid experiment validation.